# The Use of Some Polyphenols in the Modulation of Muscle Damage and Inflammation Induced by Physical Exercise: A Review

**DOI:** 10.3390/foods12050916

**Published:** 2023-02-21

**Authors:** Andressa Roehrig Volpe-Fix, Elias de França, Jean Carlos Silvestre, Ronaldo Vagner Thomatieli-Santos

**Affiliations:** 1Postgraduate Program in Psychobiology, Universidade Federal de São Paulo, Sao Paulo 05508-070, Brazil; 2Interdisciplinary Postgraduate Program in Health Sciences, Universidade Federal de São Paulo, Sao Paulo 05508-070, Brazil; 3Campus Rosinha Viegas, Universidade Metropolitana de Santos, Santos 11045-002, Brazil; 4Center for Applied Social Sciences, Universidade Católica de Santos, Santos 11015-002, Brazil

**Keywords:** flavonoids, physical exercise, inflammation, muscle damage

## Abstract

Food bioactive compounds (FBC) comprise a vast class of substances, including polyphenols, with different chemical structures, and they exert physiological effects on individuals who consume them, such as antioxidant and anti-inflammatory action. The primary food sources of the compounds are fruits, vegetables, wines, teas, seasonings, and spices, and there are still no daily recommendations for their intake. Depending on the intensity and volume, physical exercise can stimulate oxidative stress and muscle inflammation to generate muscle recovery. However, little is known about the role that polyphenols may have in the process of injury, inflammation, and muscle regeneration. This review aimed to relate the effects of supplementation with mentation with some polyphenols in oxidative stress and post-exercise inflammatory markers. The consulted papers suggest that supplementation with 74 to 900 mg of cocoa, 250 to 1000 mg of green tea extract for around 4 weeks, and 90 mg for up to 5 days of curcumin can attenuate cell damage and inflammation of stress markers of oxidative stress during and after exercise. However, regarding anthocyanins, quercetins, and resveratrol, the results are conflicting. Based on these findings, the new reflection that was made is the possible impact of supplementation associating several FBCs simultaneously. Finally, the benefits discussed here do not consider the existing divergences in the literature. Some contradictions are inherent in the few studies carried out so far. Methodological limitations, such as supplementation time, doses used, forms of supplementation, different exercise protocols, and collection times, create barriers to knowledge consolidation and must be overcome.

## 1. Introduction

The use of plant-based or natural supplements has been growing, as has the research on their properties [1]. The benefits of these supplements seem to be linked to the presence of food bioactive compounds (FBC) in the composition of leaves, roots, seeds, fungi, or seaweed [2]. FBCs are substances related to the secondary metabolism of plants, and its function is to protect against environmental aggressions [3]. There are several types of FBCs, with a huge variety of functions, with polyphenols being the most abundant [4]. It is known that FBCs bring benefits to the organism that consumes them [5], and due to the anti-inflammatory, antioxidant, and immunoregulatory [6] capacities demonstrated by some of these compounds, they have been investigated in the context of physical exercise. 

The practice of physical exercise is considered a potent immunomodulator. Beyond protecting the body against pathogens, the immune system also plays a crucial role in tissue remodeling after injury [7]. Physical exercise can induce significant injuries in the skeletal muscle tissue, leading to a consequent drop in performance. For the immune system to rescue muscle performance, improvement in pro-anti/inflammatory balance must occur to allow for muscle regeneration. However, muscle regeneration takes time [8], and hypothetically, the magnitude of cell injury (during exercise and the inflammatory phase) can be too much and make the recovery phase difficult [8]. Immunonutrition strategies to counter these deleterious effects on the immune system have been proposed [6]. Here, we discuss the role of some polyphenol supplementation strategies in muscle damage, inflammatory profile, and recovery, followed by exhaustive physical exercise. 

## 2. Food Bioactive Compounds (FBC)

FBCs comprise an immense class of substances with different chemical structures, and they exert physiological effects on the individuals who consume them [5]. They are secondary metabolites synthesized as a defense mechanism against environmental aggressions and are present mostly in plants and vegetables but also in bacteria, fungi, and, in lower concentration, animal products [3]. The positive physiological effects of the compounds are related to protection against chronic non-communicable diseases (NCDs), especially cardiovascular diseases and cancer. The negative effects are related to toxicity or allergenic potential, depending on the dose and bioavailability of the substance [5,9]. 

The main food sources of these compounds are fruits, vegetables, wines, teas, seasonings, and spices, and there are still no daily recommendations for their intake [4,10]. There are 11 major groups of compounds, classified according to their chemical structure and functionality, which may vary according to the origin of the organism. The groups are: (1) polyunsaturated fatty acids; (2) alkaloids; (3) peptides; (4) polyphenols; (5) polysaccharides; (6) triterpenes; (7) terpenoids all from plants, bacteria, fungi, and animals; (8) capsaicinoids and (9) phytosterols, produced exclusively by plants; (10) carotenoids and tocopherols, available in plants and fungi; and finally, (11) glucosinolates, synthesized by plants and bacteria [4].

Polyphenols constitute a comprehensive class of compounds with more than 8000 compounds identified [11]. Its chemical structure corresponds to one or more phenolic rings linked to hydroxyl groups [12] and is divided into flavonoids and non-flavonoids [4,10,13], as we can see in Figure 1. The direct antioxidant action (neutralization of free radicals) or indirect action (improvement of antioxidant capacity) are some highlights of the physiological effects of polyphenols [14]. In vitro studies also demonstrate anti-inflammatory activity and immunoregulatory properties [15,16,17]. As the absorption of polyphenols is conditioned on the health constitution of the intestinal microbiota and the bioavailability of these compounds varies widely, the effects demonstrated in vitro are questioned in humans [6].

Flavonoids are the most abundant and studied class [18]. They are glycosylated from polyphenols, and their best-known physiological effect is antioxidant, due to their chemical structure and to the degree of glycosylation of the molecule [19]. This class of compounds is responsible for the colorings red, blue, orange, and purplish [20], and its functions within the plant kingdom involve protection against pathogens and ultraviolet radiation [10]. The general chemical structure of flavonoids can be observed in Figure 2 and corresponds to a skeleton of 15 carbons containing two aromatic rings and a heterocyclic ring [10,21]. From this basic structure, there have derived some variations that differentiate the flavonoids in the following subclasses: flavanols, flavonols, anthocyanins, flavones, and isoflavones [22]. 

The non-flavonoid polyphenols are phenolic acids, stilbenes, and lignans [4,13]. Regarding stilbenes, resveratrol is studied for functions involving the immune system, neural protection, antitumor and antitumoral effects, and especially anti-inflammatory and antioxidant effects [23,24,25]. The main food sources of resveratrol are grapes, purple fruits and vegetables, and peanuts [13].

Although their best-known positive effects are related to NCDs, the different kinds of compounds that are studied within the scope of physical exercise involve sports performance [26,27,28], fatigue [29,30], muscle recovery [31,32,33], and immunomodulatory, antioxidant [5,6,34], and anti-inflammatory actions [32,34]. The results of studies associating polyphenol (mainly flavonoid) supplementation with exercise, especially those involving outcomes related to the immune system, are still controversial. This review relates the effects of some polyphenol supplementation—cocoa flavonols, anthocyanins, green tea catechins, curcumin, quercetin, and resveratrol—on muscle damage, inflammatory profile, and recovery, followed by exhaustive physical exercise or training. 

In this review, we do not address phenolic acid compounds and their role in performance and recovery because they were brilliantly addressed in the article by Gonçalves et al. [35].

## 3. Exercise and Their Immunomodulation Effect

During physical exercise, both in trained and sedentary individuals, it is possible to observe a brief increase in the number of circulating leukocytes, which are mobilized from the lymphatic system, vessel walls, and spleen, indicating the ability of exercise to influence different cell compartments [36]. 

In healthy people, moderate training seems to be associated with improved immunity from the point of view of antigen recognition, presentation, and elimination mechanisms, in addition to the organization of the immune response, protecting or attenuating the symptoms of infections and reducing the days with symptoms in case of illness [37,38].

On the other hand, strenuous exercise for prolonged periods has the opposite effect. After running a marathon, a picture of immunosuppression is generated, characterized by a marked decrease in the number of T cells, circulating NK cells, and neutrophils, a decrease in their activities and functions, in addition to a decrease in the salivary concentration of IgA [36]. 

In the above context, the theory of the ‘Open Window’ period after performing strenuous exercise was postulated. The ‘Open Window’ period is related to the moment after exhausting exercises that can last from 3 h to 72 h depending on the parameters analyzed, during which a lower functionality of the immune system is observed, increasing the risk and probability of opportunistic infections, mainly upper respiratory tract infections (URTIs) [39]. Performing several acute exercise sessions with strenuous characteristics without adequate recovery time can result in chronic immunosuppression. 

The modulatory effect that exercise poses on the immune system can be explained by an S-curve graphic model. Therefore, people undergoing moderate training are less likely to develop infections, especially URTIs, while amateur athletes are more likely to develop infections than people who train moderately or are sedentary. On the other hand, professional athletes, despite the high training overload, are less likely than amateur athletes to develop these infections [40,41].

The intensity and duration of physical exercise are determining factors for different changes in the immune system. Studies show that moderate aerobic training (30 to 60 min, 3 to 5 days a week at an intensity between 60 and 80% of VO_2_ max), as opposed to other intensities and volumes, results in an improvement of the immune system in the face of inflammation, in the capacity of phagocytic activity of neutrophils and monocytes [36,39,42] and a lower risk of URTIs [40]. The guiding mechanisms of exercise immunology are still discussed; however, hormonal and metabolic changes induced by exercise seem to play a relevant role [43,44,45,46,47].

### Immune System and Muscle Recovery after Exercise

The immune system plays a crucial role in tissue remodeling after injury. The adaptation of skeletal muscle tissue in response to physical exercise depends on the immune system’s function. It has been postulated that muscle adaptation/regeneration depends on the inflammatory response in a coordinated (five waves/phases) and time-dependent process [7].

The practice of physical exercise can induce significant injuries in the skeletal muscle tissue, leading to a consequent drop in performance, i.e., loss of muscle functionality. Resolving the muscle injury is paramount to restoring muscle performance. For this, during and after physical exercise, metabolites are released from damaged muscle tissue (such as CK, LDH, troponin, and complement C4) that act as damage-associated molecular patterns (DAMPs). DAMPs (the first wave) trigger an inflammatory response (i.e., recruiting immune system cells such as neutrophils, monocytes, and CD8^+^ T cells from the other sites to remove myofiber debris in the injured areas) [48]. In the second wave, neutrophils secrete IL-1 and IL-8 (which activate M1 macrophages to the lesioned region). M1 macrophages infiltrate skeletal muscle tissue to phagocytose cellular debris and secrete significant amounts of pro-inflammatory cytokines (TNF-α, IL-6, and IL-1β) and nitric oxide for a proper inflammatory response [48]. During the second wave, infiltrated CD8^+^ T cells also secrete TNF-α, IFN-γ, IL-1α, and IL-13. Pro-inflammatory cytokines such as IL1-β and TNF-α stimulate IL-6 and COX-2, which are essential to induce myoblast proliferation and differentiation (i.e., the stimulation of the myogenesis process mediated mainly by prostaglandins (PGEs and PGDs)) [49]. In the third wave, Treg cells (in response to elevated IL-6) secrete IL-33, and IL-10 stimulates the phenotypic shift from inflammatory M1 macrophages to anti-inflammatory M2 macrophages (which secrete IGF-1, IL-4, and IL-13) [48]. Anti-inflammatory cytokines (such as IL-4, IL-10, IL-13, and IL-33) provide a favorable environment for growth factors (such as IGF-1 and TGF-β) to promote the recruitment of satellite cells (to injury areas) and the differentiation, growth, and maturation of new muscle fibers [49]. Complete muscle recovery (and possible overcompensation) occurs with the maturation (fourth wave) of new muscle fibers [7]. The muscle injury/regeneration process is summarized in Figure 3.

The first wave (DAMPS) appears not to be necessary to induce better muscle adaptation [50]. On the other hand, CK secretion above normal levels has been associated with poor physical performance in athletes [46] and clinical patients [51]. Elevated CK in blood plasma occurs mainly after unaccustomed exercises or exercise protocols with weightlifting, eccentric exercises, downhill running, and prolonged exercise (e.g., ultramarathons). Well-trained individuals or those accustomed to repeated exercise to induce muscle damage showed lower muscle CK release into the bloodstream and a lower performance decrement than untrained or unaccustomed individuals [52]. Interestingly, supplementation strategies (e.g., mainly with FBC antioxidant and anti-inflammatory characteristics) decrease muscle damage after physical exercise that induces muscle damage [53]. Additionally, it is well known that FBCs increase athletic performance, such as creatine, taurine, citrulline, and nitrate, and also decrease muscle damage through antioxidant and anti-inflammatory mechanisms [53]. Therefore, preventing muscle damage with FBCs is used in sports nutrition to avoid declining athletic performance.

As described earlier, the second wave (inflammation) responds to the first wave (muscle damage) to promote the removal of damaged cell debris and to induce tissue regeneration. As indicated in Figure 2, muscle damage associated with the inflammatory process (second wave) has also been associated with decreased performance in athletes [54] and clinical patients [51]. For instance, in an experimental study, an acute IL-6 injection impaired endurance performance in healthy subjects and increased fatigue sensation [55]. Additionally, evidence has shown that the acute use of paracetamol (an inhibitor of COX-1, COX-2, and IL-6) increases athletic performance [56]. Therefore, managing inflammation-related processes might be the main reason for using sports-related pharmacological anti-inflammatory drugs because the major reason for using pharmacological anti-inflammatory drugs in athletes is to treat pain or injury, to treat illness, and to enhance performance [57]. However, the chronic and indiscriminate use of pharmacological anti-inflammatories (which interfere with the second wave described in Figure 3) can hinder the adaptation to training such as muscle strength and hypertrophy [58].

Physical exercise breaks body homeostasis, and restoring the broken balance depends on the ability of different physiological systems and cellular biochemistry to act in coordination. In this context, for exercise immunology, there is an ambiguous scenario in which metabolic, hormonal, and cytokine changes alter cell traffic directing immune system cells to skeletal muscle, despite the risk of increasing the likelihood of opportunistic infections [3,59,60]. Probably, the greater the muscle damage, the longer the cell mobilization time in the skeletal muscle and the greater the vulnerability in other body sites [8]. Therefore, strategies that can accelerate the muscle regeneration process and maintain greater control of the pro/anti-inflammatory balance after exercise can contribute to the faster restoration of muscle and immunological homeostasis. Immunonutrition strategies to counter these deleterious effects on the immune system have been proposed [6]. Following these, we report several derived FBC supplements that can influence the process of tissue repair and muscle recovery after exercise.

## 4. Cocoa

Cocoa is rich in flavanols and receives particular attention both for its palatability with good oral acceptance [61] and for its chemical composition that meets a good proportion of catechins, epicatechins, and gallocatechin [62], giving this food an interesting antioxidant potential [63]. The studies selected in this review used cocoa “*in natura*” supplementation, either beverage or chocolate. Physical exercise leads to an increase in the production of ROS, and the antioxidant system, whether endogenously represented by antioxidant enzymes or exogenously represented by the consumption of antioxidants, is responsible for maintaining the balance between the production and neutralization of ROS. Excess ROS leads to oxidative stress and inflammation, compromising performance and hindering the adaptation to exercise.

Regarding performance, Patel, Brouner, and Spendiff [64] found a 13% increase in distance traveled in a time trial test after 14 days of commercial dark chocolate supplementation (rich in (-) epicatechin) when compared to the placebo, white chocolate, both 40 g in trained men [64]. Using 7 days of supplementation (beverage enriched with flavonols—308 mg, from cocoa powder or a placebo—0 mg of flavonols) and undergoing muscle injury on day 5, trained men were evaluated for performance through vertical jump and yo-yo tests, and no differences were observed between groups [65]. The studies that evaluated the acute supplementation of cocoa flavonols, in different doses and forms of presentation, did not demonstrate differences in terms of performance. Davison et al. used 100 g of commercial 70% cocoa chocolate 2 h prior cycling for 2.5 h at ~60%VO_2_ max or a control bar in healthy men without differences in the distance [66]. Using a beverage fortified with cocoa (350 mg of flavonols, coming from cocoa powder) × a placebo drink (0 mg of flavonols), in trained men, also acutely, Peschek and Pritchett [67] did not verify differences in the performance of the time trial test nor in the strength isometric between groups. In this study, participants underwent a muscle injury induction protocol and were evaluated in a time trial test and muscle function test 48 h after injury induction [67]. In the study by Decroix and Tonoli [68], they also used a beverage as a form of cocoa supplementation (900 mg flavonols from food × 15 mg in placebo). Participants underwent a time trial test 1.5 h and 3 h after consuming the supplementation or placebo, and no performance differences were observed between groups [68].

Some studies, as seen in Table 1, evaluated the effect of CF supplementation or a placebo on the antioxidant capacity of individuals submitted to exercise. Acute supplementation (1.5 h or 3 h before exercise) of 900 mg CF from cocoa powder in a drink versus 15 mg CF in the placebo drink in trained men improved the total antioxidant capacity of plasma after a time trial test [68]. This result was also observed with a lower dose of CF (247 mg) in untrained men, who consumed 100 g of dark chocolate 2 h before low-intensity aerobic exercise [66]. The improvement of the total antioxidant capacity of plasma after physical exercise was also observed after the supplementation of 40 g of dark chocolate containing 85% cocoa for 30 days in trained men who did not receive the chocolate [69]. On the other hand, a longer time of supplementation seemed to have an opposite effect. The use of 5 g of cocoa dissolved in semi-skimmed milk (425 mg CF) for 10 weeks compared to 5 g of maltodextrin inhibited the formation of ROS (evaluated indirectly by decreasing lipid peroxidation and increasing the SOD activity), which led to lower mitochondrial biogenesis; that is, it negatively impacted the adaptation inherent to exercise [70]. The ROS and reactive nitrogen species (RNS) production, as an effect of physical exercise, is required during the inflammatory process for optimal muscle adaptation [7,28,71]. Decreasing the production of ROS with chronic CF seems to inhibit the aerobic exercise adaptation process [70]. 

Inflammatory processes resulting from the practice of physical exercise can lead to the development of muscle pain, compromising the recovery of athletes [95]. This process also depends on the magnitude of the damage caused by training [96]. Thus, modulating inflammatory cytokines and contributing to the reduction of muscle damage are important strategies for post-exercise recovery. Decreased IL-6 release (inflammatory characteristic) was observed after aerobic exercise in trained men who received 425 mg of CF (5 g of cocoa powder to be dissolved in semi-skimmed milk) for 10 weeks compared to the placebo (5 g of maltodextrin) [70]. For untrained men who received a lower dose of CF (74 mg) from cocoa juice for 7 days and underwent resistance exercises on day 8, no differences were observed in IL-6 and hsPCR levels compared to the placebo that received a drink that did not contain flavonoids [72].

Thirty days supplementation with 40 g of dark chocolate containing 85% cocoa (799 µg GAE/mL CF) in elite soccer players had a protective effect on muscle damage. Compared to the placebo group that did not received chocolate, lower levels of creatine kinase (CK) and lactate dehydrogenase (LDH) were observed [69]. Additionally, in trained individuals, the acute supplementation of a beverage containing 350 mg of CF used by Peschek and Pritchett [67] showed no changes in CK levels and no difference in muscle pain after aerobic exercise compared to the placebo, which consumed a similar beverage without CF [67]. Using another presentation (cocoa juice containing 74 mg of CF) for 10 days in untrained individuals, Morgan and Wollman [72] observed an improvement in the recovery of explosive strength after exercise compared to the placebo.

As a food, cocoa contains several FBCs (such as fatty acids, vitamins, minerals, fiber and alkaloids) [95], and the synergy between these compounds may be responsible for the observed effects. We must be careful in claiming that the results are just due to the cocoa flavonols, as these were not supplemented isolated.

## 5. Anthocyanins

Within the flavonoid group, anthocyanins are the most abundant class [97]. They are the blue, purple, red, and orange pigments that can be found in several fruits and vegetables, among which the most studied are strawberry, cherry, and blackberry [98]. The main physiological effect of anthocyanins is the improvement of endothelial function and oxidative stress, inhibiting COX-1 and COX-2 enzymes [99]. For this reason, the application of anthocyanins has been studied in diseases such as hypertension and dyslipidemia [100,101].

From the point of view of physical exercise, this effect of anthocyanins could increase performance as verified in the following works, which can be found in Table 1. The chronic use (3.3 weeks) of 300 mg anthocyanins by trained women in the form of blackberry juice—300 mg—resulted in a decrease of the time trial test time compared to the placebo [102]. In this same public, also using chronic supplementation (6 weeks) of 100 mg anthocyanins pills, an increase in VO_2_ maximum was observed in relation to placebo lactose pills [103]. A shorter time trial test was observed in trained men using 257 mg of anthocyanins, from Montmorency cherry capsules for 7 days compared to the placebo [104]. The supplementation of 116 mg of anthocyanins from a commercial juice for 9 days was able to improve the VO_2_ max of a trained man in a downhill exercise (−15%) for 30 min compared to an isocaloric maltodextrin solution [73].

Regarding muscle pain, the work of Lima et al. [73], captured in Table 1, was conducted with trained men who consumed 240 mL of an antioxidant juice twice a day (116 mg anthocyanins) for 9 days, including the day of the test, resulting in a lower perception of muscle pain using a visual scale, as well as lower CK levels evaluated 48 h and 96 h after exercise compared to the placebo group. The work of Drummer et al. [74] that evaluated muscle pain in this same public did not identify differences in this sense. In this study, participants consumed 320 mg of anthocyanins from Montmorency cherry juice or an anthocyanin-free beverage and performed an acute bout of resistance training. No differences were observed in muscle pain assessed by a visual analog scale (VAS) and pressure pain threshold (PPT). Additionally, no differences were observed in IL-6 secretion between groups [74].

Previously, the anti-inflammatory effect of anthocyanins was demonstrated in vitro. Strawberry and mulberry alcoholic extracts were added to rat splenocyte culture and the ratio of pro-inflammatory (IFN-g, IL-2 e IL-12) and anti-inflammatory (IL-10) cytokine secretion was decreased both in the presence and absence of lipopolysaccharides (LPS). In the presence of LPS, there was also a decrease in the ratio TNF-a/IL-10 [15]. It is important to highlight that the use of anthocyanins, as well as other flavonoids, is dependent on the intestinal microbiota, and therefore, there is a low bioavailability of these compounds in the human body (less than 2%). The proportion of anthocyanins eliminated is much higher than the levels of anthocyanins consumed [105,106]. This fact may explain why in vitro results do not reproduce in vivo.

In another in vitro analysis, treatment with strawberry extract decreased the ROS formation by stimulating the antioxidant enzyme activity in a RAW.7 macrophage culture LPS-challenged [17]. The effect of improving the antioxidant capacity was also observed in untrained men who consumed 500 g of strawberries in their usual diet for 14 days [16].

## 6. Green Tea (*Cammelia sinensis*)

Green tea is a very popular drink, especially in eastern countries. Therefore, the studies selected in this review used the *Cammelia sinensis* extract supplementation and not the green tea isolated FBC. Its use is widespread in sports due to its benefits in body composition, performance, recovery, and antioxidant action [107].

Catechins are the main flavonoids of green tea. The main catechins of green tea are: epigallocatechin gallate (EGCG, 60%), epigallocatechin (EG, 20%), epicatechin-3- gallate (ECG, 14%), and epicatechin (EC, 6%) [108]. This chemical structure favors the antioxidant action of green tea [109]. Although catechins play a major role in this function, green tea contains other FBCs (such as caffeine and carotenoids), chlorophyll, amino acids, and lipids [108] that may contribute to the observed effects of exercise. The interaction between these FBCs must be considered in the final analysis of the results.

In Table 1, we can see that the improvement of the total antioxidant capacity of plasma was verified in some studies. Trained men who used 250 mg of green tea extract in capsules for 4 weeks (200 mg catechins), or placebo, performed an aerobic test and observed an improvement of the total antioxidant capacity and a decrease in malondialdehyde (MDA) levels [77]. The decrease in MDA levels after the use of green tea was observed in well-trained men (elite football athletes) who maintained their usual physical activities for 6 weeks and used 450 mg of extract of green tea supplementation compared to a placebo [76].

The same authors investigated the CK levels of the athletes after the supplementation use and did not observe significant differences in comparison with the placebo. The chronic use of green tea extract in trained men did not have benefits for muscle damage indexes compared to the placebo [76] (see Table 1). In untrained men, the results are conflicting. Jajtner et al. [75] used 1 g of green tea extract supplementation (containing 50 to 80 mg of epigallocatechin gallate (EGCG)) for 28 days (or a placebo) and verified an increase in CK levels 24 h, 48 h, and 96 h after a strength training session. The use of 250 mg of green tea extract for 4 weeks following an aerobic test [77] or a 15-day supplementation of 500 mg of green tea extract following a strength exercise protocol until exhaustion [79] led to a decrease in CK concentration after exercise. Da Silva et al. [79] observed that supplementation with green tea did not alter the feeling of muscle pain reported by the participants.

The anti-inflammatory effect of green tea after physical exercise was demonstrated in overweight women who consumed 500 mg of green tea extract (225 mg EGCG) or a placebo for 8 weeks and followed an aerobic training program for the duration of the study. In this case, a decrease in hsPCR levels and an increase in plasma adiponectin levels were observed [80]. The anti-inflammatory effect was also demonstrated in untrained men who consumed 1 g of green tea extract supplementation (50–80 mg EGCG) for 28 days or a placebo and performed a strength exercise session. Intramuscular IL-8 levels were lower in the group that used supplementation compared to the placebo [75].

## 7. Curcumin

Turmeric, discovered about two centuries ago, is a tuberous herbaceous plant with yellow flowers and broad leaves that grows in tropical climates. Curcumin is an FBC derived from polyphenols and is present in turmeric; this phytochemical is responsible for the yellowish color [110,111].

Over time, studies initially demonstrated antibacterial properties [112]; however, other studies in the coming years demonstrated that curcumin has antioxidant and anti-inflammatory properties [113,114]. Thus, the body of work was growing, demonstrating the beneficial properties of this FBC in inflammatory diseases such as Alzheimer’s, Parkinson’s, rheumatoid arthritis, and diabetes [115,116,117,118].

According to the current state of the art, curcumin can contribute with its antioxidant property through phenolic groups (OH−) linked to the aromatic hydrocarbon groups in its chemical structure. These phenolic groups are essential for ROS scavenging activity [110,111]. Still, other works demonstrate that curcumin can act in the modulation of the anti-inflammatory response through different cytokines mediated by ROS, and this has been demonstrated by the reduction of the expression of the nuclear factor kappa B (NF-kB) induced by cyclooxygenase-2 (COX-2) through the inflammatory process, as well as by increasing the enzymatic antioxidant capacity through the increased activity of erythroid-derived nuclear factor 2 (NrF2) [84,119]. It is important to remember that acute physical exercise increases the NF-kB, NrF2, and COX-2 downstream canonical pathway activity, and all these molecule pathways are necessary to induce beneficial muscular adaptations [49,71]. Therefore, identifying the benefit of curcumin supplementation concomitant with physical training remains an open debate.

In the study of McFarlin et al. [81], the authors found a lower increase in CK (−69%), as well as TNF-α (−23%) and IL-8 (−23%), after the supplementation of capsules containing 400 mg of curcumin (1 ×/day) for six days in resistance training subjects compared to the placebo (see Table 1) [81]. Interestingly, using the acute supplementation of similar doses (450 mg pre-exercise) from a commercial powder dissolved in water, Mallard et al. [85] identified lower concentrations of blood lactate immediately after resistance exercise in healthy men, bringing a possible role, still poorly understood, of curcumin as an BC buffer, acting in the modulation of muscular acidosis. However, it is noteworthy that the authors did not find significant responses to other physiological markers, such as CK, LDH, and TNF-α [85]. Other studies demonstrated the benefit of curcumin supplementation at lower CK concentrations (~30% lower) after an exercise protocol to induce muscle damage in men through a 28-day supplementation but with higher doses (1.5 g of curcumin extract per day divided into three 500 mg capsules per day) [120].

According to some authors, curcumin supplementation seems not only to decrease the course of symptoms related to inflammatory responses, such as DOMS, but also contributes to attenuating the magnitude of cellular damage induced by resistance exercise. Tanabe et al. [83] identified that DOMS decreased after the supplementation of 180 mg of curcumin extract daily in 3 to 6 days after resistance exercise (see Table 1). Mallard et al. [85] observed that the acute supplementation of 450 mg of curcumin extract (30 min before resistance exercise) was able to decrease DOMS 48 h and 72 h after exercise, despite an IL-6 increase. This could be explained by an IL-10 increase and a modulation of inflammation. Authors also observed a reduction in thigh circumference in the curcumin group 24 and 48 h post exercise. This finding could suggest that a return to training may be possible earlier in the supplementation group, improving training adaptations and exercise performance [85].

A supplementation with 150 mg of a commercial curcumin extract 1 h after and 12 h before the exercise that induces loss of muscle strength could attenuate cell injuries in untrained individuals. Lower CK concentrations were observed 72 h and 96 h after exercise in the curcumin group in comparison to the placebo group. No differences were observed in neither TNF-alfa and IL-6 concentrations nor in muscle soreness (VAS) [82].

At the same time, other studies worked to demonstrate the possible antioxidant and anti-inflammatory response of curcumin supplementation (see Table 1). Takahashi et al. [86] investigated the antioxidant effects of a commercial curcumin extract in two different models: (1) a single dose 2 h before an aerobic exercise and (2) a double supplementation, 90 mg 2 h before exercise and 90 mg immediately after the same test. The single and double supplementation of curcumin could attenuate serum concentrations of derivatives of reactive oxygen metabolites (d-ROMS) and plasma thioredoxin-1 (TRX-1) and could improve the biological antioxidant potential (BAP) after exercise. Single supplementation increased the reduced glutathione (GSH) concentration after exercise. Taken together, these findings suggest that curcumin supplementation may attenuate exercise-induced oxidative stress markers by increasing antioxidant capacity [86].

A higher dose of curcumin in capsules (5 g/day) was also used by untrained men undergoing a training session that induced muscle damage. Capsules were consumed 2.5 days before and 2.5 days after exercise. Curcumin supplementation led to a decrease in DOMS assessed by VAS 24 h and 48 h after exercise. The IL-6 concentrations, which could contribute to muscle pain, had a peculiar course; they increased immediately and 48 h after exercise, but they were lower in the 24 h after the test in the supplementation group. CK activity was also lower 24 and 48 h after exercise in the curcumin group, suggesting that supplementation may attenuate muscle damage [87].

However, these findings on curcumin and the clinical trials seem to be inconclusive due to the discrepancy of studies related to its low availability and chemical instability. Apparently, higher doses of supplementation allow better effects related to polyphenol activity [121]. This low availability is due, for example, to the fact that curcumin is lipophilic, and when administered without a specific active ingredient, its absorption and, consequently, the results found may be compromised. Interestingly, in the study by Mallard et al. [85], the authors paid attention to this detail and offered curcumin extract supplementation together with 50 mg of the delivery system, LipiSperse, an active lipophilic ingredient.

## 8. Quercetin

Quercetin is a flavonol found in foods such as apples, onions, peppers, kale, pears, and spinach, among others. Quercetin can help the body with antioxidant and immunoprotective properties, modulating antioxidant responses as well as the production of cytokines [88,89,91]. Thus, in conjunction with current work, quercetin supplementation is considered a strategy to modulate the immune response of athletes by the International Olympic Committee (IOC) consensus statement [122] and can be an additional strategy to improve muscle recovery.

One of the recent studies that investigated the effects of quercetin supplementation on muscle recovery was the study by Bazzucchi et al. [89], where the authors offered 1 g of quercetin in capsules (or a placebo) for 14 days to physically active adult men and induced muscle damage via a resistance exercise bout. The authors identified decreases in CK and LDH in the quercetin group compared to the placebo immediately after and 72 h after rest [89]. Using the same supplementation and exercise protocol, now in untrained men, the same authors also identified, in 2020, that the decrease in CK and LDH seemed to remain lower for a longer period (~96 h post-exercise) (see Table 1) [90].

Furthermore, more recent studies have shown that the same amount of quercetin mentioned above, for the same period and using the same exercise protocol, seems to modulate cell injury markers for a longer period. The CK concentrations are lower in the quercetin group 72 h and 96 h post-exercise in comparison to the placebo group. LDH levels are lower in the quercetin group 48 h, 72 h, 96 h, and 7 days post exercise. The quercetin supplementation also modulates the inflammation generated by physical exercise, demonstrated by the decrease in IL-6 concentrations 48 h and 72 h post-exercise (see Table 1). All these findings indicate interesting responses to quercetin supplementation and suggest that quercetin may be used to promote a fast recovery after eccentric exercise [91].

Quercetin supplementation is also studied because its possible antioxidant effect can modulate the increase in ROS production and, therefore, prevent damage to lipid membranes, as well as protein disruption and DNA changes [123]. This antioxidant effect of quercetin supplementation was investigated by Duranti et al. [88]. In a double-blind, placebo-controlled, randomized crossover design, the authors used 1 g/d of quercetin supplementation or a placebo for 14 days in adult male volunteers, who performed maximal lengthening contractions of the upper limb at the isokinetic dynamometer following supplementation. Blood samples were collected immediately before the supplementation (0 wk), after the quercetin/placebo supplementation (2 wks), and immediately after exercise (2 wks post). In comparison with the placebo after exercise, quercetin supplementation improved the ratio of reduced/oxidized glutathione (GSH/GSSG) and decreased thiobarbituric acid (TBARS) levels, both in erythrocytes and plasma, suggesting that the chronic use of quercetin may improve redox status after a single bout of exercise [88].

On the other hand, other studies do not seem to generate scientific support of the real capacity of quercetin supplementation to beneficially modulate cell injury responses, as well as immunomodulatory responses. Nieman et al. [124] found no significant differences for IL-8 and IL-10 with 1 g of quercetin supplementation for 24 days on 40 trained male cyclists after cycling for 3 h at a 57% maximal work rate (in Watts). Additionally, O’Fallon et al. [125] demonstrated that eccentric exercise of the elbow flexors induces a small increase in plasma IL-6 ~8 h after exercise that returns to the baseline by 24 h post-exercise and that no significant differences were identified for CK, IL-6, and C-reactive protein (CRP) after supplementation with 1 g of quercetin for 7 days in 30 recreationally active subjects (15 women and 15 men).

It still seems quite complex to understand the real role of quercetin supplementation in physical exercise with the aim of modulating muscle recovery. According to studies, for immunomodulatory and antioxidant responses to occur, quercetin supplementation needs a more chronic strategy (>2 weeks) and should preferably be used in conjunction with other FBCs concomitantly [126,127]. In this sense, studies that explore the use of various FBCs such as cocoa, quercetin, and curcumin, among others, may bring better responses related to the antioxidant and anti-inflammatory modulatory capacity in practitioners of physical exercise and athletes [128,129].

## 9. Resveratrol 

Resveratrol is a stilbene often found in grapes, both internally and externally, and has been studied over the years as an interesting strategy for FBCs that brings health benefits to human bodies. Studies point out that the properties of resveratrol can help modulate chronic non-transmissible diseases by improving insulin sensitivity, altering the intestinal microbiota, as well as acting in other functions, such as modulating oxidative stress, inflammation, neurodegeneration processes, and several others [128,129]. 

Some recent works have investigated the use of resveratrol, mainly through the ingestion of grape juice, on the effects of physical exercise. Martins et al. [94] identified that 400 mL/day of grape juice for 14 days in 12 male volleyball players could help reduce lipid peroxidation and damage caused to DNA compared to a physical exercise session capable of inducing cellular damage to the muscles (see Table 1). Similar responses were found in another study, with 20 young Judo athletes, who consumed the same 400 mL/day of grape juice for 14 days; the athletes performed a Kimono Grip Strength Test (maximum number of repetitions while holding the judogi), and after the tests, lower lipid peroxidation, lower DNA damage, and lower activity of the enzyme superoxide dismutase (SOD) were identified for the group that consumed grape juice (see Table 1) [93]. Interestingly, the works cited above indicate a potential antioxidant effect of resveratrol ingestion through grape juice, which may contribute to the prevention of oxidation of lipids, proteins, and DNA (demonstrated through some blood indicators, such as malondialdehyde (MDA) and carbonyls). 

However, it is important to emphasize that this beneficial antioxidant effect of resveratrol on physical exercise is still much discussed and controversial since the works also indicate harmful effects of resveratrol intake, since this load of antioxidants can impair the adaptations that physical exercise of long duration or high intensity is capable of generating in the organism, for example, in the production of enzymatic antioxidants (catalase, superoxide dismutase, glutathione peroxidase). This impairment in adaptations was identified in the same work by Goulart et al. [93]; the authors found a lower catalase enzyme activity after the ingestion of 400 mL of grape juice. Previously, Scribbans et al. [92] also identified a lower SOD2 gene transcription in 16 recreationally active men who consumed 150 mg of resveratrol supplementation for 28 days (see Table 1). In addition, it is important to emphasize that, in grape juice, we also have the presence of phenolic acids, proanthocyanidins, anthocyanins, and flavonols, in addition to resveratrol, bringing a future discussion about studies that do not cite these compounds and their relevance in the antioxidant or anti-inflammatory system [55].

Still, several other studies have not found significant differences with resveratrol intake, indicating that the path to nutritional prescription is still quite controversial and uncertain [130,131,132].

## 10. Conclusions and Future Directions

It is classic information that exercise can result in microlesions in the exercised muscle. These injuries are the basis for post-exercise inflammation and muscle recovery, including the activation of satellite cells and the emergence of new myotubules and new cells. In this context, the muscles are more adapted to exercise.

Despite the importance of post-exercise inflammation, a high level of microdamage during exercise can be detrimental to performance, as it reflects worsening muscle function. Furthermore, a more intense and prolonged inflammatory process can partially delay the complete process of muscle regeneration. Therefore, strategies that can attenuate the injury induced by exercise and the post-exercise inflammatory process can be beneficial to accelerate recovery and help adapt to exercise.

In this context, polyphenols can have a significant nutritional impact thanks to their possible immunometabolic actions. However, discussions about the impact of polyphenols on muscle injury and post-exercise recovery are still incipient.

Due to the limited number of studies, the range of studies that used an acute exercise session to induce muscle damage and studies in which participants underwent a training period was opened. Future research should clearly distinguish effects on performance such as time trial tests or on muscle recovery, including measures of muscle regeneration and the action of inflammatory cells involved in muscle regeneration.

However, this narrative review sheds light on the role of FBCs, especially polyphenols, in post-exercise oxidative stress, injury, and inflammation. Several consulted studies show that supplementation with cocoa, anthocyanins, and quercetin can increase total antioxidant capacity and attenuate markers of injury, oxidative stress, and cytokines such as IL-6. Studies with green tea and resveratrol supplementation showed an increase in total antioxidant capacity, decreased CK and oxidative stress markers, increased total antioxidant capacity, and reduced MDA, but it could not alter cytokines. Curcumin mitigated the increase in CK and LDH and reduced TNF-α.

On the other hand, there are intriguing results and questions to be answered. For example, resveratrol supplementation reduced SOD and lipid peroxidation. However, no studies showed the impact on markers of cell damage and inflammation, despite the power that ROS has to promote the damage and modulation of inflammatory pathways, such as those regulated by NFKB. Future research should point to the role of supplementation of different types of polyphenols together and isolated since the results are impacted if the FBC supplementation is isolated (as observed in curcumin, quercetin, and resveratrol) or considering the synergy between all the FBC present in the food (as observed in cocoa, green tea, grape juice, or the antioxidant juices used in the studies discussed in the anthocyanins section).

Considering the existing variety of polyphenols, as well as their different functions, it is known that other compounds from this class not addressed in this review can affect muscle recovery [35]. Future studies could be conducted in this sense to contribute to a broad knowledge of this class of FBC in the muscle recovery process.

Some phenolic acids may have an anti-inflammatory role and consequently play a role in post-exercise inflammation and muscle recovery, including the modulatory role in intestinal microbiota and the regulation of the immune system. The relationship of the intestinal microbiota and immune system cells is based on cell regulation mediated by short-chain fatty acids and metabolites produced by the microbiota, such as butyrate. Consequently, the pro/anti-inflammatory balance may undergo adjustments based on the production of cytokines by cells in the intestinal tissue. Furthermore, phenolic acids could contribute to maintaining the integrity of the barrier function and lower LPS extravasation into the bloodstream after exhaustive exercise and recovery.

A scheme with the possible impact of joint supplements is shown in Figure 4. Collective action may more efficiently reduce oxidative stress and consequent cell damage caused by exercise. The consequence, in this case, would be less disruption of the pro/anti-inflammatory balance and less post-exercise inflammation.

The benefits discussed here do not consider the existing divergences in the literature. Some contradictions are inherent in the few studies carried out so far. Methodological limitations, such as supplementation time, doses used, forms of supplementation, different exercise protocols, and collection times, create barriers to knowledge consolidation and must be overcome.

## Figures and Tables

**Figure 1 foods-12-00916-f001:**
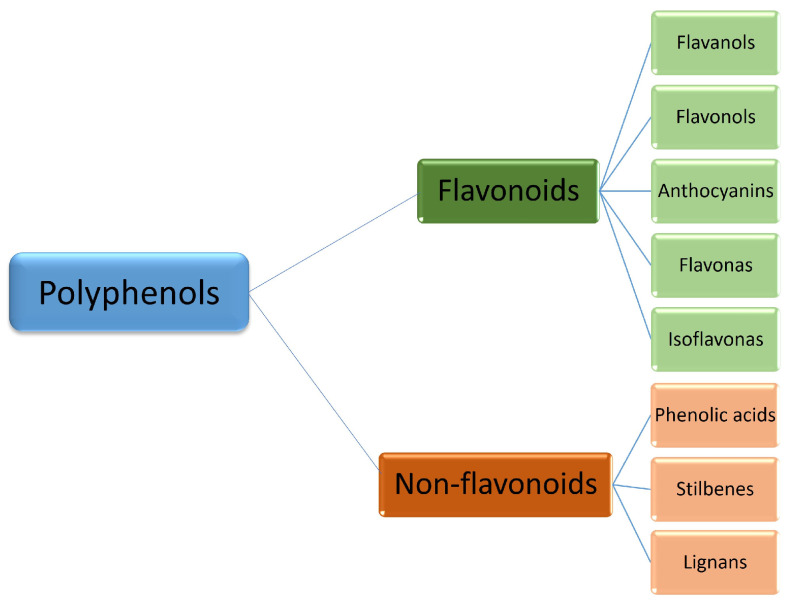
Classification of polyphenols. Adapted from Inchingolo et al. [13]; Câmara et al., 2020 [4].

**Figure 2 foods-12-00916-f002:**
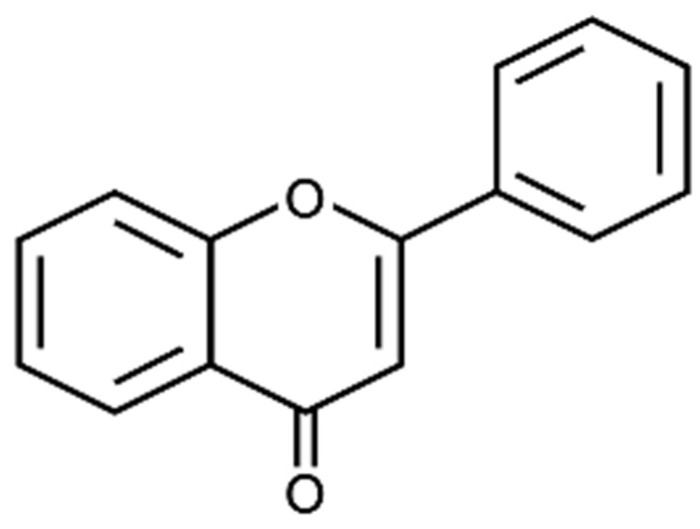
Chemical structure of flavonoids. Adapted from Nabavi et al. [21].

**Figure 3 foods-12-00916-f003:**
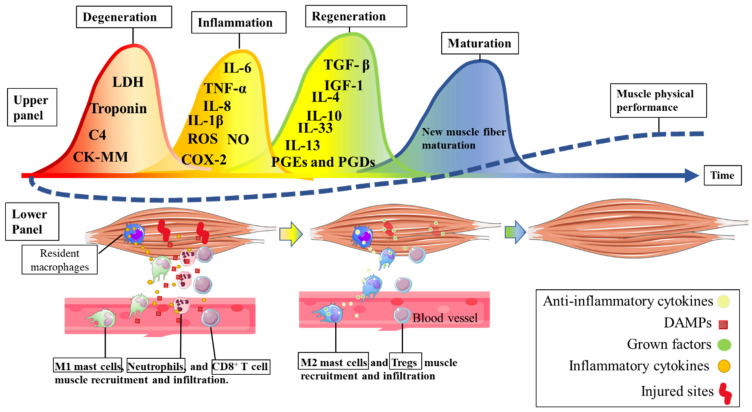
The immunological response to a strenuous bout of physical exercise. In the upper panel, the different processes (five waves) induced by muscle injury during and after physical exercise are described. The lower panel describes immune cell recruitment after muscle cell injury. Acronyms: Complement C4, C4; CK, creatine kinase; LDH, lactate dehydrogenase; COX-2, ciclooxigenase-2; IGF1, insulin grow factor-1; IL, interleukin; PGE and PGD, prostaglandins E and D, respectively; TGF-β, transforming growth factor-beta; TNF, tumor necrosis factor.

**Figure 4 foods-12-00916-f004:**
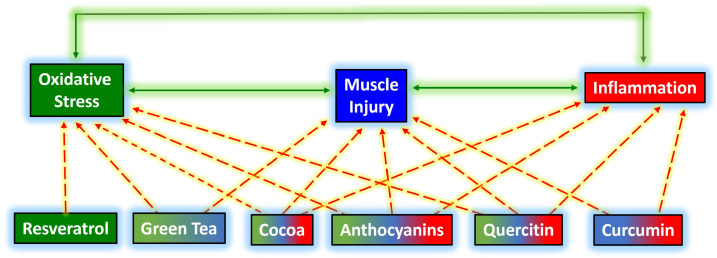
Flavonoid interaction with muscle injury, oxidative stress, and inflammation after exercise-induced muscle damage. Green solid lines signal communication between oxidative stress, muscle injury, and inflammation. The dotted red lines indicate the possible effects of different flavonoids.

**Table 1 foods-12-00916-t001:** The effect of oral polyphenol supplementation associated with physical exercise/training on IS.

Reference	Subjects	Polyphenol Type and Dosage	Exercise/Training Protocol	Immunological Outcome
De Carvalho et al. [65]	13 trained males (rugby players), age 20.6 ± 1.49 y; height 180.0 ± 0.05 cm; weight 87.02 ± 8.03 kg	308 mg of cocoa flavonols in a recovery beverage with milk (twice a day) for 7 days (total: 616 mg cocoa flavonols/d)	5 sets of 20 drop jumps from a height of 0.6 m with a 10 s break between each jump and 2 min of recovery between each set	No differences observed in performance or oxidative stress
Decroix et al. [68]	12 trained males, age 30 ± 3 y; height 177.9 ± 8.8 cm; weight 72.8 ± 7.8 kg	300 mL of a milk beverage with 900 mg of cocoa flavonols only after exercise (1.5 h before exercise)	30 min time trial cycling + 100 min rest + 30 min time trial cycling	↑ Total plasma antioxidant capacity No differences observed in time of time trial test
Garcia-Merino et al. [70]	42 trained males, age 35.18 ± 7.13; height 177.1 ± 5.84 cm; weight 71.9 ± 7.9 kg	5 g of cocoa powder dissolved in semi-skimmed milk (once a day) (total: 425 mg of cocoa flavonols/d) for 10 weeks	Exercise test in treadmill until exhaustion; then, a 1 km run 15 min after the exercise test	↓ Increment of plasma TBARS levels ↓ Increment of plasma SOD activity ↓ Increment IL-6
Peschek et al. [67]	8 trained males, 24.6 ± 5.6 y; height 182.1 ± 6.3 cm; weight 73.4 ± 7 kg	240 mL of a beverage with 350 mg of cocoa flavonols immediately after and 2 h after exercise	30 min of downhill running in a treadmill + 5 km time trial run after 48 h	No differences in 5 km time trial No differences in CK No differences in muscle soreness
Cavaretta et al. [69]	24 trained males (elite football players), age 17.2 ± 0.7 y	40 g dark chocolate (85% cocoa) (once a day) (DC) for 30 days	120 min football training 6 times per week and one 90 min match per week	↓ CK ↓ LDH ↑ Antioxidant enzymes capacity
Davison et al. [66]	14 healthy males, age 22 ±1 y; weight 71.6 ± 1.6 kg	100 g of dark chocolate (once a day) (DC) (246.8 mg CF) 2 h prior exercise bout	2.5 h cycling (~60% VO_2_ max)	↑ Insulin concentration and better glucose response ↑ Plasma total antioxidant capacity
Morgan et al. (2018) [72]	10 healthy males, age 22.8 ± 3 y; height 1.84 ± 0.59 m; weight 85.3 ± 12 kg	330 mL (once a day) of cacao juice (74 mg CF) for 10 days	3 × 3 s single leg knee extension (isocinetic dinamometer) + 3 × CMJ + 10 sets of 10 repetitions at 80% concentric 1 RM on day 8 of supplementation	↑ Recovery of explosive power
Lima et al. [73]	30 healthy males, age 22.3 ± 2.6 y; height 176.6 ± 6,4 cm; weight 77.1 ± 10.5 kg	240 mL (twice a day) of antioxidant juice (each dose with 58 mg anthocyanins. Total: 116 mg anthocyanins/d), 4 days prior, the day of, and 4 days following downhill test	Downhill running (−15%) for 30 min at 70% VO_2_ max	↓ MS ↓ CK ↑ Isometric peak torque ↑ Muscle recovery
Drummer et al. [74]	7 trained males, age 22.9 ± 4.1 y; height 180 ± 0.1 cm; weight 81.7 ± 13.2 kg	30 mL (twice a day) of concentrated Montmorrency cherry juice (each shot with 320 mg anthocyanins; total: 640 mg anthocyanins/d) for 10 days	1 bout of unilateral resistance exercise	No differences in MS, IL-6 secretion, or monocyte subset responses
Gasparrini et al. [17]	In vitro (RAW 264.7 macrophagues)	Strawberry extract (final concentration 100 µg/mL) in different concentrations	-	↑ Antioxidant defenses modulated the antioxidant enzymes (GPx, GR, GST) ↑ Inflammatory response reduced NF-kB, pIkBα, TNF-α, IL-1β, IL-6, and iNOS
Jajtner et al. [75]	38 untrained males, age 21.8 ± 2.5 y; height 171.2 ± 5.5 cm; weight 71.2 ± 8.2 kg	1 g (twice a day) of oral Camelia sinensis extract in a capsule for 28 days (each capsule with 50–80 mg de EGCG; total: 2 g polyphenol/day)	6 sets of 10 repetitions of squats; 4 sets of 10 repetitions of the leg press and leg extension	↑ CK ↓ IL-8 intramuscular
Hadi et al. [76]	54 trained males (soccer players), age 29.9 ± 1.43 y; height 180.88 ± 6.06 cm; weight 74.12 ± 8.62 kg	450 mg (once a day) of oral Camelia sinensis extract in capsules for 6 weeks	Continued with regular soccer trainning	↓ MDA levels No differences in muscle damage indices
Chi-Kuo et al. [77]	40 healthy males, age 20 ± 1 y; height 172 ± 5.5 cm; weight 66 ± 8.1 kg	250 mg (once a day) of oral Camelia sinensis extract in capsules for 4 weeks	20 min moderate-intensity cycling (75% VO_2_ max) 3 × week	↑ Run time to exaustion ↑ Total antioxidant capacity ↓ CK
Jówko et al. [78]	16 trained males, age 21.6 ± 1.5 y; height 180.5 ± 6.2 cm; weight 76.9 ± 6.4 kg	250 mg (2 capsules, twice a day) of oral Camelia sinensis extract in capsules for 4 weeks (each capsule with 245 mg polyphenols—200 mg catechins and 137 mg EGC-3-galate. Total: 980 mg polyphenols/day)	2 repeated cycle sprints (4 × 15 s, with 1 min rest intervals)	↑ Total antioxidant capacity at rest ↓ MDA levels after exercise
Da Silva et al. [79]	20 untrained males, age 25 ± 5 y; height 173 ± 6 cm; weight 76 ± 9 kg	500 mg (once a day) of oral Camelia sinensis extract in capsules for 15 days	Calf raising until exhaustion	↓ CK No effects in delayed onset muscle soreness sensation
Bagheri et al. [80]	30 overweight females, age 38.3 ± 3.16 y	500 mg (once a day) of oral Camelia sinensis extract in capsules for 8 weeks	Endurance training 3 x/week	↓ Body weight, body mass index, waist to hip ratio and fat percentage ↓hs-PCR ↑ Adiponectin
McFarlin et al. [81]	28 healthy males and females, age 20 ± 2 y; height 168 ± 9 cm; body fat 24 ± 12.1%	400 mg (once a day) of oral Curcumin supplementation for 6 days	6 sets of 10 repetitions of the leg press exercise with a beginning load set at 110% of their estimated 1 RM	Curcumin Group: 2 days after: ↓~18% IL-8 4 days after: ↓~69% CK ↓~23% TNF-α
Tanabe et al. [82]	14 untrained young men, age 24 ± 1 y; height 172.1 ± 7.5 cm; weight 65.2 ± 11.3 kg	300 mg (150 mg 1 h before and 150 mg 12 h after eccentric exercise) of oral Curcumin supplementation	50 maximal eccentric contractions of the elbow flexors	Curcumin Group: ↓~56% CK 96 h post-exercise
Tanabe et al. [83]	10 healthy males, age 29 ± 3.9 y; height 172.6 ± 5.1 cm; body fat 20 ± 5.6%	90 mg (twice a day) of oral Curcumin supplementation for 7 days	30 maximal eccentric contractions of the elbow	Curcumin Group: ↓ VAS 3, 4, 5. and 6 days post-exercise ↓ CK 5, 6, and 7 days post-exercise
Basham et al. [84]	20 healthy males, age 21.7 ± 2.9 y; height 177.7 ± 7.4 cm; weight 83.7 ± 12.4 kg	1.5 g (once a day) of oral Curcumin supplementation for 28 days	15 min of continuous sitting with one leg at a height of 42 cm using an aerobic step bench with a cadence of 15 sit-stand repetitions/min for a total of 225 repetitions	Curcumin Group: ↓~30% CK post-exercise
Mallard et al. [85]	27 trained males, age 26 ± 5 y; height 184 ± 7 cm; weight 86.42 ±10.8 kg	450 mg (once a day) of oral Curcumin extract 30 min after exercise	4 sets of leg press at 80% 1 RM until exhaustion	↓ MS 48 h and 72 h post-exercise ↓ Thigh circumference 24 h and 48 h and post-exercise ↓ Lactate concentration ↑ IL-10 e IL-6
Takahashi et al. [86]	10 healthy males, age 26.8 ± 2 y; height 173 ± 5 cm; weight 67.7 ± 5.3 kg	90 mg of oral Curcumin extract 2 h before exercise OR 90 mg of oral Curcumin extract 2 h before and immediately after exercise	Walk (or run) at 65% VO_2_ max for 60 min	↓ d-ROMS and TRX-1 ↑ Biological antioxidant potential (BAP) ↑ Reduced glutathione (GSH)
Nicol et al. [87]	17 healthy males, age 33.8 ± 5.4 y; weight 83.9 ± 10 kg	2.5 g (twice a day) of oral Curcumin extract 2.5 days prior and 2.5 after exercise	7 sets of 10 eccentric single-leg press repetitions	↓ MS 24 and 48 h post-exercise ↓ CK activity 24 and 48 h post-exercise
Duranti et al. [88]	14 men, age 25.5 ± 0.8 y; height 179 ± 10 cm; BMI 23.4 ± 0.5 kg/m^2^	2 capsules containing 500 mg (twice a day) of Quercetin supplementation for 14 days	Completing maximal lengthening contractions of the upper limb at the isokinetic dynamometer	Quercetin Group: post-exercise ↓ 30.3% GSSG ↑33.9%GSH/GSSG ↓ 31.9% TBARS
Bazzucchi et al. [89]	20 young men, age 26.1 ± 3.1 y; height 179 ± 4 cm; weight 75.1 ± 7.1 kg	500 mg at breakfast and 500 mg 12 h later of oral capsules of Quercetin supplementation for 14 days	10 bouts of 10 maximal lengthening contractions of the elbow flexors; each set was separated by a 30 s rest	Quercetin Group: ↓ 19.6% LDH 24 h post-exercise ↓ 12.5% LDH 48 h post-exercise ↓ 17.5% LDH 72 h post-exercise ↓ 52.7% CK 48 h post-exercise ↓ 78.6% CK 72 h post-exercise
Bazzucchi et al. [90]	16 men, age 25.9 ± 3.1 y; BMI 23.4 ± 2.0 kg/m^2^	2 capsules containing 500 mg (twice a day) of Quercetin supplementation for 14 days	10 bouts of 10 maximal lengthening contractions of the elbow flexors; each set was separated by a 30 s rest	Quercetin Group: ↓ LDH 48, 72, and 96 h post-exercise ↓ CK 72 and 96 h post-exercise
Sgrò et al. [91]	12 healthy moderately active young men, age 25.6 ± 3.8 y; weight 77.4 ± 5.11 kg; BMI 24.5 ± 1.3 kg/m^2^	500 mg at breakfast and 500 mg 12 h later of oral capsules of Quercetin supplementation for 14 days	10 bouts of 10 maximal lengthening contractions of the elbow flexors; each set was separated by a 30 s rest	Quercetin Group: ↓ CK 72 and 96 h post-exercise ↓ LDH 24, 48, 72, and 96 h, and 7 days’ post-exercise ↓ IL-6 48 and 72 h post-exercise
Scribbans et al. [92]	16 recreationally active men, age~22 ± 1 y; VO_2_ max~51 mL/kg/min	Pills of 150 mg of oral Resveratrol supplementation 15 min after exercise (training days) OR breakfast (nonexercised days) for 28 days	Eight 20 s intervals at 170% of peak aerobic work rate (WRpeak) separated by 10 s of rest. Three training sessions were completed per week over the 4 week intervention period	Resveratrol Group: ↓ PGC-1α mRNA ↓ SIRT1 mRNA ↓ SOD2 mRNA
Goulart et al. [93]	20 Judo athletes, age 17.8 ± 2.2 y; height 165 ± 0.1 cm; body fat 13.8 ± 8%	Drink 400 mL (once a day) of grape juice for 14 days	Kimono Grip Strength Test (maximum number of repetitions while holding the judogi)	Grape Juice Group: ↓ 10% Lipid Peroxidation ↓ 19% DNA damage ↓ 80% SOD activity
Martins et al. [94]	12 male volleyball players, age 16 ± 0.6 y; height 186.6 cm ± 8.41 cm; body fat 14 ± 3.3%	Drink 400 mL (once a day) of grape juice for 14 days	Handgrip strength was evaluated using a hydraulic dynamometer. The athletes performed three repetitions in each hand (right and left) with an interval of 60 s between them	Grape Juice Group: ↓ Lipid Peroxidation ↓ DNA damage

Acronyms: ↑, increase; ↓, decrease; CK, creatine kinase; IFN, interferon; IL, interleukin; LDH, lactate dehydrogenase; TNF-α, tumor necrose factor alpha; VAS, visual analog scale; TBARS, thiobarbituric acid reactive substances; SOD, superoxide dismutase; EGCG, epigallocatechin-3-gallate; TT, time trial; MS, muscle soreness; MDA, malondialdehyde; hs-PCR, high sensitivity C-reactive protein; d-ROMS, derivatives of reactive oxygen metabolites; TRX-1, plasma thioredoxin-1; CF, cocoa flavonols; GPx, glutathione peroxidase; GST, glutathione S-transferase; GSSG, oxidized glutathione; GSH, reduced glutathione; GR, glutathione reductase; PGC-1α, peroxisome proliferator-activated receptor-gamma coactivator; SIRT1, sirtuína 1; iNOS, inducible nitric oxide synthetase; NF-kB, nuclear factor kappa B; pIkBα, phospho-inhibitory subunit of NF-KBα.

## Data Availability

Data sharing is not applicable to this article.

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
