# Peer review of "The Use of Some Polyphenols in the Modulation of Muscle Damage and Inflammation Induced by Physical Exercise: A Review"

_foods, 2023, doi:10.3390/foods12050916_

Round 1
Reviewer 1 Report
I find the review interesting, and it is an area that has not been sufficiently explored. However, it would be better if the authors should make a clear distinction between isolated compounds (curcumin, resveratrol, etc.) with those that are a mixture of compounds (cocoa, green tea, etc.). This is relevant since the effects of a mixture of compounds cannot be compared with one isolated. It is also important to note that several papers study the effect of (-)-epicatechin on muscle. None of these studies are mentioned in the manuscript. I think it would also be advisable to mention the application routes of the products since this often determines the secondary metabolites, which are the ones that have the effect.
Author Response
Thanks to the reviewer for important and pertinent suggestions for improving the manuscript. Below we answer all the questions and suggestions. We are also sending attached the paper with all the accepted suggestions.
Referee: “... it would be better if the authors should make a clear distinction between isolated compounds (curcumin, resveratrol, etc.) with those that are a mixture of compounds (cocoa, green tea, etc.). This is relevant since the effects of a mixture of compounds cannot be compared with one isolated.”
Response: Inclusion of the distinction suggested in section 4 – Cocoa (lines 238 - 242), section 6 – Green tea (lines 369 - 371) and conclusion (lines 601 - 603).
Referee: “I think it would also be advisable to mention the application routes of the products since this often determines the secondary metabolites, which are the ones that have the effect.”
Response: We have mentioned the routes of ingestion as well as the frequency of supplementation in Table 1 for ease of visualization.
Referee: “It is also important to note that several papers study the effect of (-)-epicatechin on muscle. None of these studies are mentioned in the manuscript.”
Response: Due to the popularity of green tea consumption, it was decided to observe the effect of whole plant and not of any bioactive compound isolated, and its included (-)-epicatechin. Justification was included in lines 369 to 371: “Therefore, the studies selected in this review used the Cammelia sinensis extract supplementation and not the green tea FBC isolated (e.g catechins).”

Reviewer 2 Report
Interesting to see a paper looking at this specific issue around muscle recovery from exercise damage
It does appear there is currently quite limited specific studies on this topic
I think you need to clarify and be consistent with language around the compounds being described or discussed, use food bioactive compounds (FBC) or related terminology, bioactive compounds is too broad.
The abstract needs to be clearer and there are some difficult sentences there, I would suggest use food bioactive compounds (FBC) as used in your Ref 4. Most the last half of abstract is a bit repetitive and could be more brief I would say.
Not all the compounds described are flavonoids in the broad sense. In particular the stilbene resveratrol and diarylheptanoid curcumin don't match the title description. Some of the categorisation of the flavonoids is mixed, and needs to be more consistent and include anthocyanins earlier. The reference 4 cited is a pretty good review with good overview of ow complex polyphenolics are.
In particular the studies involving Grape juice need to refer also to the presence of phenolic acids, flavanols, procyanidins, flavone glycosides and anthocyanins as much as resveratrol. Resveratrol often occurs at far lower levels in Grape juice than these other phenolics.
Whilst they are not directly covered here there are phenolic acids that likely have related anti-inflammatory activity and also may impact sugar and fat metabolism, gut microbiota and immune function, which might also affect muscle metabolism and recovery.
While the overall role of inflammation and immune function in response to exercise, muscle damage and recovery seems good. There also seems to be a mix in the studies between effects on exercise or muscle performance and or post exercise recovery or inflammatory markers. I realise there is probably limited current and specific studies in this. However the relationship between improved or enhanced performance and muscle damage and recovery is not clear, I think even from the studies cited. As I believe you point out in brief.
I would think a review could highlight these issues and try to describe what might be the most useful in future research to distinguish performance measures with repeat exercises or specifically look at this issue of muscle damage and recovery.
Some of this is discussed very briefly but seems like this could be discussed more directly.
The dosage and length of treatment or supplementation, including some requirement to quantify the levels of FBC's in the supplement or dosage taken, whether pre and post exercise, and issues around absorption of different compounds is another issue. Given that curcumin is not a very water soluble compound and not well absorbed on it's own, I was a bit sceptical about it's potential role in the results described in the studies cited, eg Mallard et al Ref 69, which was described as a water based curcumin supplement. Curcumin being lipophilic is also not so readily absorbed on its own. Perhaps some evaluation of the studies you cite and their direct relevance and quality in regard to muscle damage and recovery would be useful for this review.
Another issue which is just touched on is the relationship between these FBC's and gut microbiota and the likely interaction between immune function and overall inflammation related to this. The connection between the gut microbiota and inflammation has been subject of much recent literature including studies looking at the role of these FBC's in altering gut microbiota.
Author Response
Thanks to the reviewer for important and pertinent suggestions for improving the manuscript. Below we answer all the questions and suggestions. We are also sending attached the paper with all the accepted suggestions.
Referee: “I think you need to clarify and be consistent with language around the compounds being described or discussed, use food bioactive compounds (FBC) or related terminology, bioactive compounds is too broad.”
Response: We have changed the terminology to the recommended one (food bioactive compounds - FBC).
Referee: “The abstract needs to be clearer and there are some difficult sentences there, I would suggest use food bioactive compounds (FBC) as used in your Ref 4. Most the last half of abstract is a bit repetitive and could be more brief I would say.”
Response: The abstract has been rewritten.
Referee: “Not all the compounds described are flavonoids in the broad sense. In particular the stilbene resveratrol and diarylheptanoid curcumin don't match the title description. Some of the categorisation of the flavonoids is mixed, and needs to be more consistent and include anthocyanins earlier. The reference 4 cited is a pretty good review with good overview of ow complex polyphenolics are.”
Response:
We have made the following changes to the manuscript:
- Changing the tittle to suit the nomenclature, as well throughout the text.
- Classification adequacy throughout the text in section 2 – Food Bioactive Compounds (FBC)
- Inclusion of Figure 1
- Inclusion of the paragraph below on page 3 (lines 102 to 106):
“The non-flavonoid polyphenols are phenolic acids, stilbenes, and lignans [4,13]. Regarding stilbenes, resveratrol is studied for functions involving the immune system, neural protection, antitumor and antitumoral effects, and especially anti-inflammatory and antioxidant effects [23-25]. The main food sources of resveratrol are grapes, purple fruits and vegetables, and peanuts [13].”
Referee: Whilst they are not directly covered here there are phenolic acids that likely have related anti-inflammatory activity and also may impact sugar and fat metabolism, gut microbiota and immune function, which might also affect muscle metabolism and recovery.
Response: We added information in the text that addresses this issue (631 - 639).
Referee: “While the overall role of inflammation and immune function in response to exercise, muscle damage and recovery seems good. There also seems to be a mix in the studies between effects on exercise or muscle performance and or post exercise recovery or inflammatory markers. I realise there is probably limited current and specific studies in this. However, the relationship between improved or enhanced performance and muscle damage and recovery is not clear, I think even from the studies cited. As I believe you point out in brief.”
Response: We've extended the topic to discuss the relationship between muscle damage, inflammation and performance (line 195 – 222)
Referee: “I would think a review could highlight these issues and try to describe what might be the most useful in future research to distinguish performance measures with repeat exercises or specifically look at this issue of muscle damage and recovery.”
Response: We insert the 604 - 608 lines to suggest possible fields that can be investigated in the future (Lines 604 - 608).
Referee: “In particular the studies involving Grape juice need to refer also to the presence of phenolic acids, flavanols, procyanidins, flavone glycosides and anthocyanins as much as resveratrol. Resveratrol often occurs at far lower levels in Grape juice than these other phenolics.”
Response: In lines 582 to 585 we include the other bioactive compounds in addition to resveratrol.
Referee: “The dosage and length of treatment or supplementation, including some requirement to quantify the levels of FBC's in the supplement or dosage taken, whether pre and post exercise, and issues around absorption of different compounds is another issue. Given that curcumin is not a very water-soluble compound and not well absorbed on it's own, I was a bit sceptical about it's potential role in the results described in the studies cited, eg Mallard et al Ref 69, which was described as a water-based curcumin supplement. Curcumin being lipophilic is also not so readily absorbed on its own. Perhaps some evaluation of the studies you cite and their direct relevance and quality in regard to muscle damage and recovery would be useful for this review.”
Response: We've also made changes to the last few paragraphs on curcumin, including discussions of availability and chemical instability. The changes are on lines 482 to 490.
Referee: Moderate English changes required.
Response: We proofread the text for possible grammatical and typing errors.

Round 2
Reviewer 1 Report
The authors responded satisfactorily to the comments
Reviewer 2 Report
Looks a much clearer and more focused review with consistent terminology, a good start in looking at what is required in future investigations